# Stabilization of three-dimensional charge order through interplanar orbital hybridization in $Pr_xY_{1-x}Ba_2Cu_3O_{6+\delta}$

Alejandro Ruiz[1,14], Brandon Gunn[1,14], Yi Lu[2,3], Kalyan Sasmal[1], Camilla M. Moir[1], Rourav Basak [1], Hai Huang [4,13], Jun-Sik Lee [4], Fanny Rodolakis [5], Timothy J. Boyle[6,7], Morgan Walker[6], Yu He[8], Santiago Blanco-Canosa[9,10], Eduardo H. da Silva Neto[6,7,11], M. Brian Maple[1] & Alex Frano [1,12] ✉

The shape of $3d$-orbitals often governs the electronic and magnetic properties of correlated transition metal oxides. In the superconducting cuprates, the planar confinement of the $d_{x^2-y^2}$ orbital dictates the two-dimensional nature of the unconventional superconductivity and a competing charge order. Achieving orbital-specific control of the electronic structure to allow coupling pathways across adjacent planes would enable direct assessment of the role of dimensionality in the intertwined orders. Using Cu $L_3$ and Pr $M_5$ resonant x-ray scattering and first-principles calculations, we report a highly correlated three-dimensional charge order in Pr-substituted $YBa_2Cu_3O_7$, where the Pr $f$-electrons create a direct orbital bridge between $CuO_2$ planes. With this we demonstrate that interplanar orbital engineering can be used to surgically control electronic phases in correlated oxides and other layered materials.

The cuprate phase diagram illustrates a quintessential example of a low-dimensional correlated quantum system: a multitude of fascinating electronic phases, including spin-density waves, charge and nematic order, and high-temperature superconductivity[1] (SC), emanating from the combination of complex interactions that govern a simple two-dimensional (2D) chemical structure − the $CuO_2$ planes. These interactions within the $CuO_2$ planes are largely influenced by the underlying characteristics of the anisotropic, planar Cu $3d_{x^2-y^2}$ orbitals, which dominate the density of states near the Fermi surface due to a sizeable energy splitting of the $e_g$ orbitals[2]. The 2D character of the surviving $d_{x^2-y^2}$ orbital is directly observable in transport measurements, evidenced by its anisotropic electrical and thermal

conductances[3], for example. The 2D character of the system is also observable in scattering measurements, where the lack of coupling pathways between adjacent planes can yield overwhelmingly broad scattering peaks along $L$ in reciprocal space[4]. Understanding how the dimensionality of the orbital degrees of freedom affect the stability and interplay of these phases could reveal important information about the mechanism of superconductivity with broader applications for modifying the characteristics of correlated oxides and layered materials via orbital engineering.

Among the most important cases of this interplay is the phenomenon of charge order (CO), a phase that is closely interconnected with SC[5–18]. Incommensurate CO exists in all superconducting cuprates

[1]Department of Physics, Center for Advanced Nanoscience, University of California, San Diego, CA 92093, USA. [2]National Laboratory of Solid State Microstructures and Department of Physics, Nanjing University, 210093 Nanjing, China. [3]Collaborative Innovation Center of Advanced Microstructures, Nanjing University, 210093 Nanjing, China. [4]Stanford Synchrotron Radiation Lightsource, SLAC National Accelerator Laboratory, Menlo Park, CA 94025, USA. [5]Advanced Photon Source, Argonne National Laboratory, Argonne, IL 60439, USA. [6]Department of Physics, University of California, Davis, CA 95616, USA. [7]Department of Physics, Yale University, New Haven, CT 06520, USA. [8]Department of Applied Physics, Yale University, New Haven, CT 06511, USA. [9]Donostia International Physics Center, DIPC, 20018 Donostia-San Sebastian, Basque Country, Spain. [10]IKERBASQUE, Basque Foundation for Science, 48013 Bilbao, Spain. [11]Energy Sciences Institute, Yale University, West Haven, CT 06516, USA. [12]Canadian Institute for Advanced Research, Toronto, ON M5G 1M1, Canada. [13]Present address: Department of Materials Science, Fudan University, 200433 Shanghai, China. [14]These authors contributed equally: Alejandro Ruiz, Brandon Gunn. ✉e-mail: afrano@ucsd.edu

as a 2D electronic phenomenon hosted in the $CuO_2$ planes, reflecting the weak interplanar coupling of the planar Cu $3d_{x^2-y^2}$ orbitals. In diffraction experiments, the 2D character is evidenced by a reciprocal space 'rod' that is broad along the out-of-plane direction (Miller index $L$), and maximized at half-integer values of $L$ due to a weak, but out-of-phase, coupling between adjacent planes[19]. The strength of this interplanar coupling can be further quantified by extracting the correlation lengths from the widths of the scattered CO peaks along $L$. In $YBa_2Cu_3O_{6+\delta}$ (YBCO), the highest reported out-of-plane correlation length (10 Å) is nearly an order of magnitude smaller than the highest reported in-plane correlation length (95 Å)[5], highlighting the 2D nature of the CO phase. It is not clear whether disorder[20–22] or the low dimensionality of the underlying Cu $3d_{x^2-y^2}$ orbitals intrinsically limits the out-of-plane correlation length, or if the CO could, in principle, develop into a truly long-range order, as suggested by recent experiments[20–22].

It has since been observed that the application of certain perturbations – high magnetic fields[4,23–25], epitaxial strain in thin films[26], or uniaxial strain[27,28] – can induce a CO phase with three-dimensional (3D) coherence. Upon the application of these external influences, a second CO peak emerges, this time centered at integer $L$-values, evidencing an out-of-plane coupling that locks the phase of adjacent $CuO_2$ planes. The 3D CO peaks have significantly increased out-of-plane correlation lengths, achieving up to 55 Å[24], 61 Å[26], and 94 Å[27], respectively. All of these 3D CO correlation lengths are still considerably shorter than the typical crystalline $c$-axis correlation lengths found in this compound. Furthermore, the 2D rod centered at half-integer $L$-values gets enhanced upon applying the external influences, showing a persistent coexistence of the 3D and 2D COs. While it is easy to discern the 2D nature of the unperturbed CO upon consideration of the underlying planar Cu $3d_{x^2-y^2}$ orbitals, the mechanisms by which these external perturbations are able to induce a 3D CO peak remain unclear. Moreover, the in situ application of these perturbations presents complicated technical challenges that preclude many experimental techniques altogether, making it difficult to systematically investigate how the dimensionality of the CO can be tuned and obscuring its connection with SC. Taking an orthogonal route, we hypothesized that 3D CO could instead be stabilized by virtue of tuning the underlying orbital character via hybridization to more directly enhance the out-of-plane coupling between adjacent $CuO_2$ plane layers.

Here we show that, by substituting Pr on the Y sites in $Pr_xY_{1-x}Ba_2Cu_3O_7$ (Pr-YBCO) (Fig. 1a), a highly correlated 3D CO state can be stabilized with an out-of-plane correlation length of ~ 364 Å (Fig. 1b), a number that is bound by the crystalline correlation length, within our experimental resolution. This material was chosen because substitution by Pr, which is the largest trivalent rare-earth ion, except for Ce which does not form the YBCO structure[29], results in the emergence of hybridization between the Pr $4f$ orbitals and planar $CuO_2$ states[30] that yields an electronically relevant, hybridized orbital[31] with spatial extension in three dimensions, in stark contrast to the planar Cu $3d_{x^2-y^2}$ orbitals that dominate the physics of the parent compound. Unlike substitution by other rare-earth elements, such as Dy, which do not significantly alter the parent YBCO phase diagram[32], increasing Pr substitution in the $Pr_xY_{1-x}Ba_2Cu_3O_7$ system continually reduces the superconducting $T_c$, yielding a pseudogap regime[30,33,34] and eventually an antiferromagnetic insulating phase[30,35–40]. Furthermore, the in- and out-of-plane zero-temperature superconducting coherence lengths are substantially longer in Pr-YBCO than in YBCO and increase monotonically with Pr concentration[41–43]. This indicates that SC gains additional 3D character with increasing Pr substitution and has been attributed to increased coupling between $CuO_2$ planes through the bonding with the substituted Pr ions. Various results suggest that localized Pr $4f$ states are appreciably hybridized with the valence band states associated with the conducting $CuO_2$ planes, specifically the O $2p$ level[44,45]. We present density-functional calculations showing that, through this hybridization which is unique to Pr, the CO on adjacent $CuO_2$ planes can couple to yield a stable 3D CO phase. Altogether, our results constitute the first detection of a fully stabilized, long-range 3D CO that competes with SC, achieved by intrinsically engineering the orbital character of the electronic structure.

## Results

We used resonant soft x-ray scattering (RSXS) at the Cu $L_3$ and Pr $M_5$ edges to investigate the CO properties in a $Pr_xY_{1-x}Ba_2Cu_3O_7$ sample with $x \approx 0.3$ and a superconducting $T_c = 50$ K; a concentration value chosen because it features pseudogap behavior, as measured by various probes[30,33,34], and because it yields a $T_c$ similar to underdoped

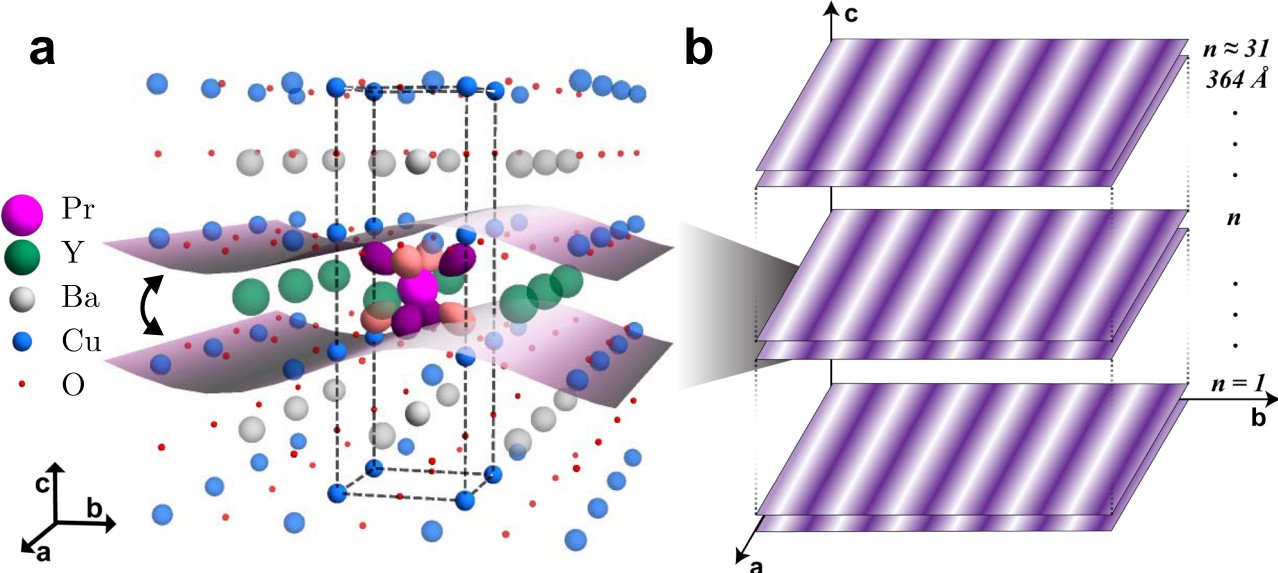

**Fig. 1 | Stabilizing three-dimensional charge order. a** The extended unit cell (dashed box) of $Pr_xY_{1-x}Ba_2Cu_3O_7$ illustrating the charge order coupling between adjacent $CuO_2$ planes that arises when introducing Pr at the Y sites. The Pr $4f_{z(x^2-y^2)}$ orbital is shown which hybridizes with the $2p_\pi$ states of the planar O. **b** A schematic depiction of the 3D CO out-of-plane correlation length in $Pr_xY_{1-x}Ba_2Cu_3O_7$ which spans ~31 sets of $CuO_2$ planes (~364 Å).

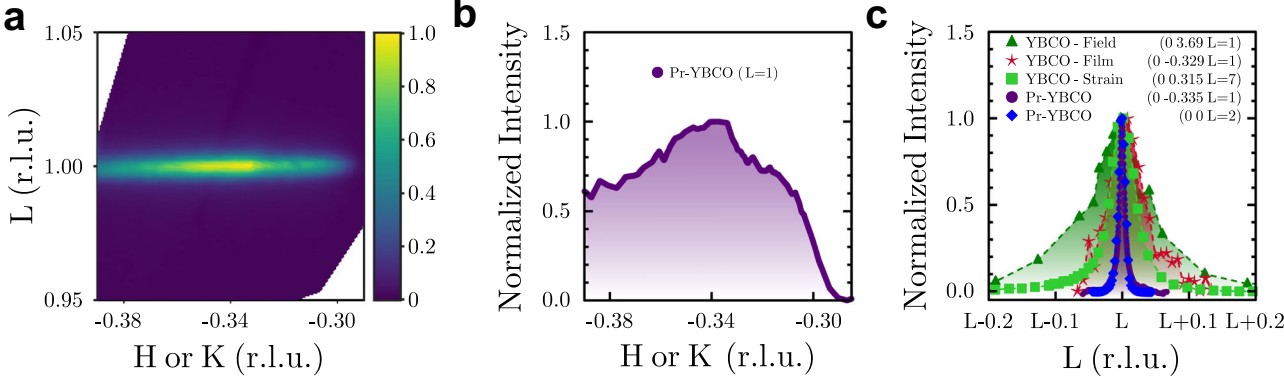

**Fig. 2 | The reciprocal space structure of the 3D CO at $T = 50$ K. a** An *HL or KL* reciprocal space map collected at 932.4 eV shows a diffraction feature centered at (0 -0.335 1) reciprocal lattice units (r.l.u.). **b** A cut along *H or K* at $L = 1$ shows the peak is centered around a value of *H or K* = -0.335 (r.l.u.). **c** A comparison of the out-of-plane *L*-widths of the known 3D CO peaks stabilized by magnetic field[4], epitaxial strain in YBCO films[26], uniaxial strain[27], and the present work. For comparison, an *L*-cut of the structural (002) peak from Pr-YBCO is shown.

$YBa_2Cu_3O_{-6.67}$, a doping level where the CO phase is maximal. Due to not having detwinned samples (Methods section), we cannot determine whether the 3D CO peak is biaxial or uniaxial. If the 3D CO is uniaxial, we cannot determine whether it is located along the *H* or *K* reciprocal axis. The location of the 3D CO peak is thereby referred to as being along *H or K* to reflect this, except for places where the position is labeled simply by *K* for the sake of readability.

### Reciprocal space dependence

A reciprocal space map of the *HL or KL*-plane in reciprocal lattice units (r.l.u.), measured at $T_c = 50$ K at 932.4 eV is shown in Fig. 2a. In eminent contrast to all other reports of 3D CO, no scattered intensity was detected in the vicinity of $L \approx 1.5$, indicating the apparent absence of 2D CO (see Supplementary Methods). This represents the first unique aspect of our work: to within the limits of our instrumental resolution, we only observe a peak at $L = 1$, suggesting an effective isolation of the CO phase with an out-of-plane coupling.

Further inspection of the 3D CO signal displays a reciprocal space structure that is broad along *H or K* but narrow along *L*. X-ray absorption fine structure measurements indicate that, while the Pr ions are relatively well-ordered at the Y sites, there is clear disorder in the $CuO_2$ planes and in the oxygen environment around the Pr[29], which makes the enhancement of the correlation length along the *c*-axis even more striking. The broad shape of the peak along *H or K* at $L = 1$, shown in Fig. 2b, is consistent with the broad feature observed in many previous RSXS measurements of CO in cuprates[12,14,15,46] that has been attributed to a fluctuating component in YBCO[47], suggesting that the actual static contribution may be narrower than it appears.

Another important feature of our discovery is shown in Fig. 2c, which compares reciprocal space cuts along *L* close to integer values with *K* centered at the in-plane CO wavevector. The broadest peak (dark green triangles) displays the data reported for 3D CO induced by high magnetic field[4]. The next broadest peak (red stars) displays the data for 3D CO induced by epitaxial strain in a thin film[26]. The next broadest peak (lime green squares) displays the data measured under the application of 1.0% uniaxial strain[27], which has yielded 3D CO with the previously highest reported out-of-plane correlation length. The Pr-YBCO 3D CO peak (purple circles) is considerably narrower, yielding a correlation length of ~364 Å. This value is found to be similar to the absorption length for this compound, photon energy (~930 eV), and angle of incidence (~10°). The observable correlation length of the 3D CO may thus be limited by the finite penetration depth being of similar magnitude. We believe this is not a significant factor, however, due to the (002) structural reflection (blue diamonds) having a correlation length that is within the experimental uncertainty of the 3D CO, even

though it was measured at higher energy (~1750 eV) and angle of incidence (~38°), both of which contribute to a significantly longer absorption length. This suggests that, in this Pr-YBCO system, the 3D CO peak has a width that is limited by the width of the crystallographic Bragg peaks. The measured ~364 Å thus represents a lower bound on the out-of-plane correlation length.

### Energy dependence

The energy dependence of the scattered intensity at $Q = (0 -0.335 1)$ is shown in Fig. 3a, overlaid with the corresponding x-ray absorption spectrum (XAS) measured with the electric field of the x-rays parallel to the bond directions in the $CuO_2$ planes. The XAS reveals two resonances that correspond to the Pr $M_5$ (930.9 eV) and the Cu $L_3$ (932.6 eV) edges. There are also two peaks observed in the energy dependence of the 3D CO (930.3 eV and 932.8 eV), which most likely correspond to contributions from Pr and Cu, respectively. However, due to the energetic overlap of the Pr $M_5$ and Cu $L_3$ edges, the energy dependence of the scattering is unavoidably complex; as such, we refer to them simply as peaks A and B (see Supplementary Discussion). Unlike in YBCO films with 3D CO[26], we do not observe a significant shift in spectral weight to higher energy that would indicate CO coupling through the CuO chains. Furthermore, we observe in Fig. 3b that the 3D CO peak can still be detected at energies far below the resonance (850 eV) albeit much more weakly, which is in contrast to all other cuprates where the CO peaks studied by RSXS lack sufficient scattering strength to be observed off-resonance. This indicates a sizeable lattice distortion rarely seen[7] in other cuprate systems, highlighting that in Pr-YBCO the 3D CO becomes more structurally stable than previously reported.

### Temperature dependence

The interplay of the 3D CO with superconductivity can be investigated by measuring the temperature dependence of the former. In Fig. 3c, we plot the scattered intensity at $Q = (0 -0.335 1)$ at energies corresponding to peaks A and B in the energy dependence as a function of temperature. While the overall scattering intensity is higher at the peak A energy than the peak B energy, which is consistent with the measured energy dependence, it is notable that the 3D CO scattering signal is still detectable at room temperature for both energies. Upon cooling from $T = 300$ K, the temperature dependence at both energies maintain roughly equivalent slopes until within the vicinity of $T_c = 50$ K. Cooling below $T_c$ produces a cusp-like maximum, indicating a competition between SC and the isolated 3D CO phase. This signature behavior confirms that CO is at least a major contributor to the observed scattering, regardless of whether additional structural

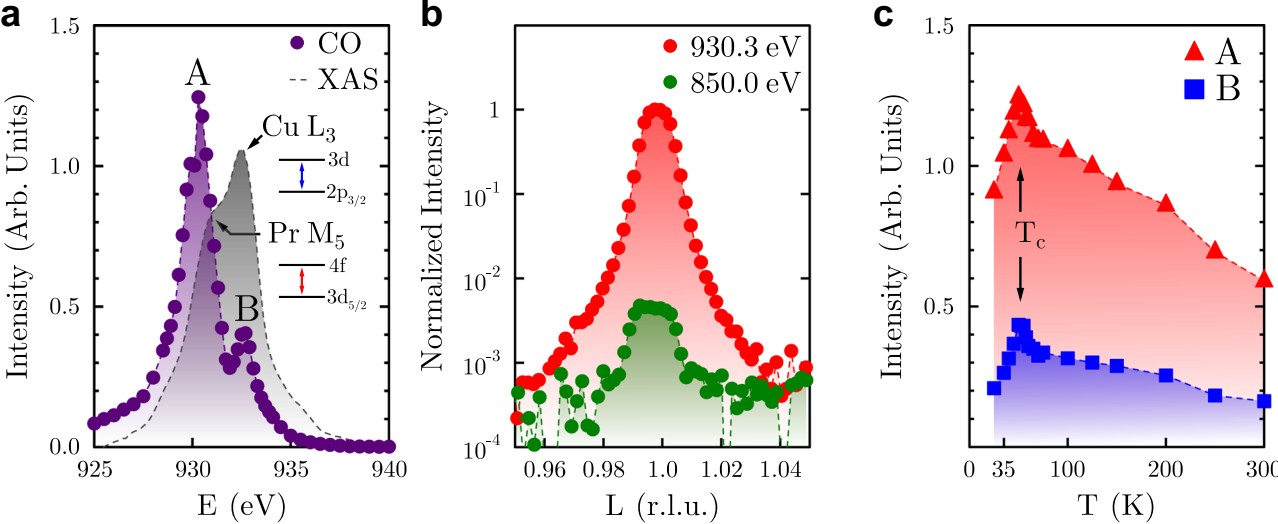

**Fig. 3 | The nature of the 3D CO. a** The x-ray absorption spectrum (XAS) (gray) shows the Cu $L_3$ edge with a shoulder at lower energies corresponding to the Pr $M_5$ edge. The two dipole-allowed transitions are labeled as insets. The purple data show the energy dependence of the scattered intensity of the 3D CO at $T_c = 50$ K with the two most prominent features labeled as peaks A and B. **b** A semi-log plot of rocking curve scans of the 3D CO peak taken on resonance (930.3 eV) and well below the resonance (850 eV); the latter displays a weaker, but detectable, off-resonant intensity. **c** The temperature dependences measured at energies corresponding to the two peaks observed in the 3D CO energy dependence (A: red triangles; B: blue squares), showing detectable peaks at room temperature and drops below the superconducting transition ($T_c$).

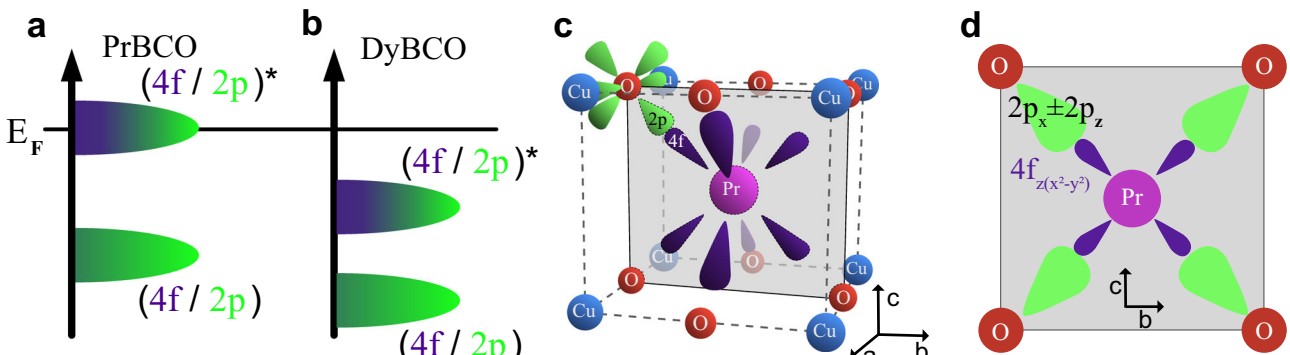

**Fig. 4 | DFT-calculated band structures showing hybridization. a** A schematic representation of the orbital character of the electronic levels near the Fermi energy ($E_F$) in PrBa$_2$Cu$_3$O$_6$ (PrBCO), showing a mixed $4f$ (purple) and $2p$ (green) antibonding band crossing $E_F$. **b** The equivalent schematic for DyBa$_2$Cu$_3$O$_6$ (DyBCO). **c** A schematic depicting the $4f$ (purple) and $2p$ (green) orbitals within the crystal structure of PrBCO in three dimensions. **d** The hybridization that occurs between the $4f_{z(x^2-y^2)}$ (purple) and a linear combination of $2p_x$ and $2p_z$ orbitals (green) oriented towards each other, shown within the plane represented in panel **c**.

contributions exist. This is in contrast to the 3D CO induced by very high magnetic fields, where any competition between 3D CO and SC is obscured by the very presence of the magnetic field which, while necessary to induce 3D coherence, comes with the unavoidable expense of greatly suppressing the SC phase.

## Discussion

Having established experimentally that 3D CO can be stabilized with long out-of-plane correlation length, we turn to discuss the possible origin of the *c*-axis coupling in the Pr-YBCO system. It is already well known that, unlike any other rare earth, Pr substitution uniquely suppresses SC in YBCO by localizing holes via orbital hybridization[44,45,48,49]. To this end, we performed density- functional theory plus Hubbard U (DFT+U) calculations for both PrBa$_2$Cu$_3$O$_6$ (PrBCO) and DyBa$_2$Cu$_3$O$_6$ (DyBCO) structures[50] to understand the role of this hybridization within the context of 3D CO and its competition with SC (see Supplementary Methods). Figure 4a schematically depicts

the orbital character of the electronic states near the Fermi level ($E_F$) in PrBCO. In addition to the characteristic $pd\sigma$ bands of the CuO$_2$ planes which host all the 2D electronic phenomena, another band crosses the Fermi level. From prior calculations[49], it is clear that the effective doping is affected, as this band above the Fermi level takes holes from the superconducting band, which is consistent with the observation that $T_c$ is suppressed with increasing Pr concentration. This results from the antibonding coupling between the Pr $4f_{z(x^2-y^2)}$ state and its nearest-neighbor O $2p_\pi$ states in adjacent CuO$_2$ planes[48,49] (Fig. 4c, d). We speculate that this orbital coupling with an out-of-plane component locks together the phase of the CO on adjacent CuO$_2$ planes, resulting in a diffraction peak at $L = 1$[19]. For later rare-earth elements with lower $4f$ energy, the $4f_{z(x^2-y^2)}-2p_\pi$ antibonding band is expected to be lowered and removed from the Fermi level. Figure 4b depicts the calculated electronic level structure for DyBCO, where the top of this band is deep below the Fermi level at around −0.8 eV. This leaves the charge carriers in the two adjacent CuO$_2$ planes essentially decoupled. This unique

aspect of Pr makes it the appropriate rare-earth to substitute into Y to stabilize and isolate 3D CO, which occurs concomitantly with a lattice distortion, according to our data. The exact structural mechanism of stabilization, e.g., phonons, is a subject of future research.

Our discovery of a fully stable 3D CO without a 2D signal has important implications to our understanding of CO and its interplay with SC. First, we confirm that a fully coherent, isolated 3D CO can be stabilized despite the intrinsic disorder inevitably present in cuprates. We note here that our Pr-substituted samples are expected to host at least as much structural and chemical disorder than in pristine YBCO, if not more so, due to the additional defect channel. This result may further elucidate the complex relationship between CO and SC, both of which have now been shown to substantially gain 3D character with increasing Pr concentration in the Pr-YBCO system[41–43]. Second, we confirm that a stable 3D CO still coexists and competes with SC, implying that the system's ground state can comprise two long-range, static, coexisting orders. Third, since the 3D coupling does not rely on the CuO chains that are unique to YBCO, perhaps other forms of hybridization can be used to stabilize 3D CO in other cuprate families, which has not yet been observed. Finally, we show that controlling the orbital content of the Fermi surface by assigning it a $4f$ character with an out-of-plane component can yield a sizable impact on the electronic ordering tendencies of the $CuO_2$ plane. It can be used as a tuning knob to study the validity of 2D models to describe layered systems, like the cuprates or intercalated graphitic systems[51–55].

In summary, we have shown how utilizing the hybridization between the $4f$ states of Pr and planar $CuO_2$ orbitals to tune the underlying orbital character can significantly enhance the out-of-plane coupling, phase-locking the CO across adjacent planes and rendering a stable CO phase that is fully correlated along the out-of-plane direction without the 2D version. The $c$-axis correlation length has a lower bound matching that of the crystal itself, showing that Pr substitution is the most efficient way of stabilizing 3D CO compared to using external perturbations, like magnetic fields and strain, and uniquely does not suffer from experimental complications arising from in situ application. Furthermore, through resonant spectroscopy, we attribute the formation of 3D coupling to the role of the Pr ions located between $CuO_2$ planes. To understand the mechanism of this out-of-plane coupling, we turned to DFT+U calculations that show a hybridized $4f$-$2p$ band crossing the Fermi level, a feature that is unique to Pr-substituted YBCO. Since our system does not rely on external perturbations, other techniques can be employed to investigate this material and shed light on the connection between CO and SC. Moreover, this demonstrates how the influence of underlying orbital character on an electronic phase can be tuned via orbital hybridization, which can be generalized to other correlated transition metal oxides and layered systems.

## Methods
### Sample preparation
Single crystals of $Pr_xY_{1-x}Ba_2Cu_3O_7$[45] were grown according to the method described in reference[56]. The starting materials used in the crystal growth consisted of 99.99% pure $Y_2O_3$, $Pr_6O_{11}$, $BaCO_3$, and CuO powders. The crystals were annealed in flowing oxygen to maintain full oxygenation and optimize their superconducting properties. The $Pr_{0.3}Y_{0.7}Ba_2Cu_3O_{-7}$ sample we studied has an orthorhombic crystal structure that is not detwinned with lattice parameters $c = 11.67$ Å and $a = b = 3.87$ Å. The superconducting transition temperatures of the crystals were determined from magnetization measurements performed with a vibrating sample magnetometer in a Quantum Design DynaCool Physical Property Measurement System.

### RSXS measurement
The data shown in this manuscript were collected from scattering experiments carried out at beam line 13-3 of the Stanford Synchrotron Radiation Lightsource (SSRL). Crucial measurements and insights were gained through scattering experiments carried out at Sector 29 of the Advanced Photon Source (APS). The sample was mounted using silver paint on an in-vacuum multiple-circle diffractometer. The sample temperature was controlled by an open-circle helium cryostat. The incident photon polarization was fixed as $\sigma$ (vertical linear) polarization. The $(0\ K\ L)$ scattering plane was determined by aligning the $(0\ 0\ 2)$, $(0\ \text{-}1\ 1)$, and $(0\ 1\ 1)$ structural Bragg reflections at 1746 eV photon energy. We note that we have also observed this phenomenon in a second sample with a similar Pr concentration (see Supplementary Note).

A $256 \times 1024$ pixel ($26\ \mu m \times 26\ \mu m$ pixel size) CCD detector was used. The scattering intensity data were collected within a region-of-interest in the center of the CCD detector. Dark images and data measured by the CCD detector outside of this region-of-interest were used to subtract any background fluorescence contributions, which were generally very small compared to the 3D CO scattered intensity, except for when off-resonance or at high temperature. A beam shutter was used to cut the incoming x-ray beam between two consecutive CCD shots to prevent undesired collection of x-ray photons during read-out. A 100 nm Parylene/100 nm Al filter was placed in front of the CCD to stop photoelectrons emitted from the sample from contributing to the signal on the CCD. Further details about the data collection and analysis methods used may be found in the Supplementary Methods.

## Data availability
The data generated in this study have been deposited in the Harvard Dataverse database available at https://doi.org/10.7910/DVN/2BIWWI.

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

## Acknowledgements

We thank Bernhard Keimer, Davide Betto, Fabio Boschini, George Sawatzky, Martin Bluschke, Matteo Minola, Mingu Kang, and Riccardo Comin for fruitful discussions. This material is based upon work supported by the National Science Foundation under Grant No.

DMR-2145080. Research at UC San Diego (M.B.M., K.S., and C.M.M.) was supported by the US Department of Energy, Office of Basic Energy Sciences, Division of Materials Sciences and Engineering, under Grant No. DE-FG02-04ER46105 (single crystal growth) and US National Science Foundation under Grant No. DMR-1810310 (materials characterization). Resonant soft x-ray experiments were carried out at the SSRL (beamline 13-3), SLAC National Accelerator Laboratory. This study at the SSRL/SLAC is supported by the U.S. Department of Energy, Office of Science, Office of Basic Energy Sciences under contract no. DE- AC02-76SF00515. This research also used resources of the Advanced Photon Source (29ID), a U.S. Department of Energy (DOE) Office of Science User Facility operated for the DOE Office of Science by Argonne National Laboratory under Contract No. DE-AC02-06CH11357. Y.L. acknowledges support by Deutsche Forschungsgemeinschaft (DFG) under Germany's Excellence Strategy EXC2181/1-390900948 (the Heidelberg STRUC-TURES Excellence Cluster). Y.H. acknowledges previous support from the Miller Institute for Basic Research in Sciences. S.B-C acknowledges support from the MINECO of Spain through the project PGC2018-101334-AC22. A.F. was supported by the Research Corporation for Science Advancement via the Cottrell Scholar Award (27551) and the CIFAR Azrieli Global Scholars program. E.H.d.S.N. acknowledges previous support from UC Davis startup funds, as well as current support from the Alfred P. Sloan Fellowship in Physics.

## Author contributions

A.F. and M.B.M. conceived and led the project. The RSXS experiments were performed by A.R., B.G., H. H., J.-S.L., F.R., T.J.B., M. W., Y.H., S.B.-C., and E.H.d.S.N. Single crystals were grown and characterized by K.S., C.M.M., and M.B.M. The data analysis was carried out by A.R., B.G., H.H., and J.-S.L. DFT+U calculations performed by Y.L. The manuscript was written by A.F., A.R., B.G., Y.H., E.H.d.S.N, and S.B.C. with input from all the co-authors.

## Competing interests

The authors declare no competing interests.
