## [Peer Review File · Nature Communications]

REVIEWER COMMENTS

Reviewer #1 (Remarks to the Author):

Ruiz et al. present resonant x-ray scattering measurement on Pr doped YBCO. They find scattering that is sharp in L at peaked at an integer value of L ($L=1$), similar to previously reported measurements of more 3D CDW order in YBCO in applied magnetic field, under uniaxial strain or in strained thin films. Moreover, they present evidence the this peak persists up to high temperature and is suppressed in intensity below the superconductivity transition temperature. While this is an intriguing result, I have several significant reservations about the technical presentation of the data and the conclusions thusly drawn. As such, I do not recommend publication of the manuscript.

Significant issues

1. The presentation of the data using rocking scans is quite misleading. For instance, the data shown figure 2B (at fixed L) is quite different from figure 3B (where L varies over the scan). In the latter case, the peak appears to be sharp in K, when in fact it may only be sharp in L. It has become common in studies of 2D CDW order to perform rocking scans (varying θ , but not the detector angle 2θ) in order to determine the width of a peak in H or K. This is reasonable in instances where the peaks are broad in L, such that the variation in peak intensity in L is small relative to variation in H or K over the theta angle range covered. However, this case appears to be the opposite limit, such that rocking curves should be plotted vs. L instead of K. Moreover, the peak in K is oddly shaped, dropping off at low K, and somewhat unlike the Lorentzian like CDW peaks that are often observed YBCO. This makes a determination of a peak value difficult and of lower confidence that the commensurate value of 0.335 claimed in the text. Were true K scans at fixed L done at other temperatures or energies. If so, a presentation of this data would be instructive.

2. The Pr and Cu edges clearly overlap in energy in this case, with the width of the Pr edge being ~ 5 eV wide. As such a clean attribution of scattering to Pr sites at 930.5 eV and Cu at 933 eV. For instance, it is conceivable that scattering at both energies is dominated by Pr atoms, with little contribution from Cu.

3. If the sample is not detwinned, then the measurements will potentially be measuring H or K or both. The authors should not label the peaks only by K.

4. An alternate explanation for some aspects of the data, including that the peaks are observed off resonance, may be diffuse scattering from Pr disorder. The nature of the disorder may be such that peak are narrow in L (with little deviation of the Pr positions from their ideal values along the c-axis), but more positional significant disorder within the ab plane. Some mention of this possibility is perhaps warranted in the paper. Measurement of another YBCO variant with other substitution for Y (say Dy) may provide insight into this possibility. Notably, the temperature dependence of the peak, particularly the dip below T_c does not fit neatly into this this description.

More minor issues

1. For resonant soft x-ray scattering of long-range order, the width of a peak may be determined by the finite penetration depth of the x-rays into the sample rather than representing the correlation length of some order. Evidence that this is the case can be a peak width that varies with photon energy (since the absorption length varies with photon energy). Can the authors compare the peak width to the expected absorption length at the measured photon energies?

2. Even off resonance, there should be a significant reduction in scattering intensity for pi polarization relative to sigma polarization for this scattering geometry. As such the presentation of the polarization dependence of the scattering intensity does not necessarily provide an indication that the states are in-plane states. This is associated with the standard polarization factor for x-ray diffraction where scattering for pi polarized photons is zero when the incident and scattered photon polarization are orthogonal. Without further analysis, it is not clear to me that the scattering involves planar orbitals.

3. The "XAS" measured in TFY is likely not a good measure of the x-ray absorption. It may be affected by self-absorption corrections at the Pr edge and will also be affected by the difference in quantum efficiency of fluorescence yield for Pr M emission vs. Cu L emission. When edges do not overlap, this can be ignored, but when they do, the FY spectra is unlikely to resemble the x-ray absorption coefficient. This should be clarified in the text.

4. How was the scattering intensity vs energy measured? This is not specified in the paper. Was this via an energy scan at fixed Q? Energy scans at fixed Q can be misleading for sharp peaks that have peak position that varies as a function of photon energy due to the energy dependence of the index of refraction.

5. In lines 43- 45 , the authors note: "The 2D character of the surviving $d_{x^2-y^2}$ orbital is observable in both transport and scattering measurements, for example, evidenced by its anisotropic electrical and thermal conductances³ and overwhelmingly broad scattering peaks along L in reciprocal space⁴, respectively."

It is understandable that transport is anisotropic in these materials, due to the dx^2-y^2 character of the states orbitals at the Fermi surface (this is found from first principles DFT calculations), I don't think the relationship between the fact that holes are in dx^2-y^2 orbitals and the correlation length of CDW peaks along the L direction is very clear. The coupling between layers rather than the orbital symmetry within an individual layer may be most relevant for the c-axis correlation length. For instance, ortho ordering peaks in YBCO are broad in L. These peaks, however, principle result from Cu and oxygen in the chain layers, which involve Cu dy^2-z^2 states and O p_z states – states are that are not orthogonal to z. As such, it is coupling between layers rather than the orbital symmetry of an individual layer that may be most relevant.

Reviewer #2 (Remarks to the Author):

The authors present a resonant soft x-ray scattering (RSXS) study of Pr-substituted YBCO and find charge order at a wavevector (0, -0.335, 1). The key and unique finding is an extremely long c-axis correlation length of at least 364 Å. This value far exceeds the correlation lengths of similar charge order reported previously in YBCO perturbed by strong magnetic fields or strain. The mechanism of the strong c-axis correlations is argued to be related to interlayer coupling mediated by the Pr 4f orbitals, which hybridize with the O 2p orbitals in the CuO₂ planes. The findings are striking and clearly presented and demonstrate control of strongly correlated phases through "orbital engineering". I am in favor of publication in Nature Communications although there are a few minor points and questions the authors should address.

1. Did the authors collect data on the temperature dependence of the lineshape? Fig 3c shows plots of intensity vs temperature (I assume this means peak intensity), but if possible, it would be useful to see the dependence of correlation length and/or integrated intensity vs temperature as well.

2. How does the in-plane correlation length of the CO compare to that in Refs. 4, 23-27? On first glance it appears the peak in Fig 2b is much wider than that in the previous references. This might be related to the increased chemical disorder due to Pr substitution, in which case the enhancement of correlation length along the c-axis is even more striking. It might be worth noting this in the text and possibly including in the supplement a comparison similar to Fig 2c, but along K.

3. lines 188-189: Single-band models can describe 3d systems, although not Pr-YBCO since there are two CuO₂ layers per unit cell. If I understand the intention of this sentence correctly, it would be more accurate to say "...validity of 2d models to describe layered systems like the cuprates...".

4. lines 138-139: "sizeable lattice distortion unseen in other cuprate system". Is this true? Tranquada Nature 375, 561–563 (1995) reported observation of charge order via neutron scattering, due to coupling of the lattice to charge. Certainly the present result is striking, given that this has not been observed previously in YBCO.

Reviewer #3 (Remarks to the Author):

I have read the manuscript "Stabilization of three-dimensional charge order CO through interplanar orbital hybridization in Pr_xY_{1-x}Ba₂Cu₃O_{6+δ}" by Alejandro Ruiz and collaborators submitted to Nature Communications for consideration. The manuscript reports resonant x ray diffraction signatures of charge order which do not commensurate with the real space lattice. Pr substitution triggers a change from 2D charge order with short out of plane correlation (10 Å in underdoped cuprates with similar oxygen content), to 3D charge order with a long out of plane correlation length of 360 Å, which is actually instrument resolution limited. Resonant measurements at the Pr M edge confirm that Pr orbitals play a role in the 3D coupling. DFT +U calculations are used to show that the Pr f_z²-y₂ band, crossing the Fermi level, is responsible for the out of plane coupling of the charge ordered states.

This is an important manuscript showing that the 3D CO developing in underdoped cuprates triggered by disorder, strain or large magnetic field is in fact a different form of static order coexisting and competing with superconductivity. Furthermore, this paper adds the important new information that the 2D CO concomitant with the superconducting state in underdoped cuprates is not an essential ingredient in the big picture of the high T_c superconductivity of the cuprates.

An important information missing is how the proposed electronic coupling through the Pr f states affects other electronic states, aside from the CO state, in particular how doping is modified (if it is at all). In addition, it is important to know how Pr doping modifies the dimensionality of the superconducting state and the extent superconductivity becomes more 3D as also does the CO state.

In addition, it would be very important to have information on the evolution of the out of plane correlation length of the CO state with doping. I.e., if hole doping of the Pr doped compound (by oxygen depletion for example) would restore the 2D picture.

Reviewer #1 (Remarks to the Author):

Ruiz et al. present resonant x-ray scattering measurement on Pr doped YBCO. They find scattering that is sharp in L at peaked at an integer value of L ($L=1$), similar to previously reported measurements of more 3D CDW order in YBCO in applied magnetic field, under uniaxial strain or in strained thin films. Moreover, they present evidence the this peak persists up to high temperature and is suppressed in intensity below the superconductivity transition temperature. While this is an intriguing result, I have several significant reservations about the technical presentation of the data and the conclusions thusly drawn. As such, I do not recommend publication of the manuscript.

Significant issues

1. The presentation of the data using rocking scans is quite misleading. For instance, the data shown figure 2B (at fixed L) is quite different from figure 3B (where L varies over th scan). In the latter case, the peak appears to be sharp in K, when in fact it may only be sharp in L. It has become common in studies of 2D CDW order to perform rocking scans (varying θ , but not the detector angle 2θ) in order to determine the width of a peak in H or K. This is reasonable in instances where the peaks are broad in L, such that the variation in peak intensity in L is small relative to variation in H or K over the theta angle range covered. However, this case appears to be the opposite limit, such that rocking curves should be plotted vs. L instead of K.

We thank the Reviewer for bringing this important point to our attention. We agree that Figure 3B could be improved, as suggested. We have replotted the rocking curves in Figure 3B as functions of L, instead of H or K, as shown below:

In an effort to make the experimental methods more comprehensible to the reader, we have additionally added the following plot and some discussion so that it is precisely known what is meant by a rocking curve scan:

FIG. S1. The trajectory of a rocking curve scan through the location of the center of the 3D CO peak. In this mode, the CCD detector angle (2θ) remains fixed while the sample angle (θ) is varied. By measuring rocking curve scans through a range of CCD detector angles (2θ), various regions of the HL - or KL -plane may be spanned.

Moreover, the peak in K is oddly shaped, dropping off at low K, and somewhat unlike the Lorentzian like CDW peaks that are often observed YBCO. This make a determination of a peak value difficult and of lower confidence that the commensurate value of 0.335 claimed in the text.

We agree with the Reviewer that an assessment of commensurability is not straightforward in this case due to the shape and broadness of the in-plane peak. We have removed any statements about the peak being nearly commensurate from the text as it does not diminish from the central message of a long out-of-plane correlation length.

Were true K scans at fixed L done at other temperatures or energies. If so, a presentation of this data would be instructive.

Measuring a true K scan at fixed L necessarily requires moving the detector at each step, which is not ideal as it can introduce issues such as artifacting. The preferable method of data collection is by measuring rocking curve scans, which allows the detector to remain stationary throughout the entirety of the scan. While a single rocking curve scan is a reasonable measure of the peak along L for the reasons that the Reviewer has mentioned, this makes measuring along K at fixed

L non-trivial and requires measuring a collection of rocking curve scans spanning a region of the KL-plane, from which the dependence on K at fixed L must be extracted. Due to time constraints, we were unable to measure the region of the KL-plane required to extract the dependence on K at fixed L at different temperatures or energies.

2. The Pr and Cu edges clearly overlap in energy in this case, with the width of the Pr edge being ~ 5 eV wide. As such a clean attribution of scattering to Pr sites at 930.5 eV and Cu at 933 eV. For instance, it is conceivable that scattering at both energies is dominated by Pr atoms, with little contribution from Cu.

This raises an important concern and we are grateful that the Reviewer has brought it to our attention. Considering the close proximity of these two absorption edges, we believe that a discussion of this point is certainly warranted in the text, which we have added. However, we maintain a high degree of confidence that the peaks in the 3D CO energy dependence are attributed accurately, with the peak ~ 933 eV corresponding to Cu and not Pr, for the reasons listed below:

Reason 1: It's the simplest explanation

The temperature dependence exhibiting a cusp-like maximum at the onset of the superconducting T_c , with reduced scattering intensity below T_c , is a hallmark of CO that has been observed in all families of cuprates (in the absence of Pr) and is believed to signify competition with superconductivity, which is believed to reside within the CuO_2 planes. While it would be quite interesting for the Pr (located at the yttrium sites) to be the only species active in the 3D CO formation while still exhibiting a temperature dependence that has the same distinct behavior as in all other cuprates that do not contain Pr, we do not see any evidence of this in the data and believe this scenario to be highly unlikely and that it is much more probable that the peak at ~ 933 eV in the 3D CO energy dependence simply corresponds to involvement by the Cu species, which reconciles why the distinct behavior of the temperature dependence parallels all other cuprates without Pr and how the competition with superconductivity in the CuO_2 planes may be preserved.

Reason 2: Similarity between the XAS and 3D CO energy dependence peak locations

We would first like to clarify that the width of the Pr edge alone is closer to ~ 2.7 eV, rather than 5 eV, as shown in the fit below:

Further evidence that the 3D CO peak at ~ 933 eV corresponds to the Cu edge is because of how closely the energy matches with the Cu L_3 absorption edge at ~ 932.6 eV, which has been measured and calibrated at this beam line (which has an energy resolution of ~ 0.1 eV near the Cu L_3 and Pr M_5 edges) many times before. It can be seen in the plot below that the maximum of the 3D CO peak is positioned approximately at 932.76 eV, which matches very closely with the Cu L_3 edge position at 932.59 eV.

We note that the 3D CO energy dependence resonating at slightly different energies than their corresponding absorption edges is consistent with other resonant scattering measurements of 3D CO in YBCO.

<https://www.nature.com/articles/s41467-018-05434-8>

Reason 3: The separation between the peaks in the 3D CO energy dependence is greater than the reported separation between Pr³⁺ and Pr⁴⁺ valence states in XAS measurements

Further evidence that the 3D CO peak ~933 eV is attributed to the Cu resonance is due to the separation of these peaks being ~2.49 eV apart. The XAS measurements from the study below find the separation to be only ~1.5 eV between trivalent and tetravalent Pr species.

FIG. 2. (Color online) From top to bottom: experimental Pr $M_{4,5}$ absorption spectra of Pr tetravalent BaPrO_3 (violet) and trivalent Pr_2O_3 (green) [both from Ref. 28], $\text{Pr}_{0.5}\text{Ca}_{0.5}\text{CoO}_3$ measured at 10 K (blue) and 300 K (red), and a comparison of Pr_2O_3 to the 85:15 weighted addition (dark blue) of Pr_2O_3 and BaPrO_3 spectra, i.e., ideally the spectrum of a compound containing in average $\text{Pr}^{+3.15}$. Some spectra have been vertically shifted for clarity reasons.

<https://journals.aps.org/prb/pdf/10.1103/PhysRevB.84.115131>

Reason 4: We do not expect significant contributions from more than one Pr valence in this material

Additionally, various spectroscopic probes have consistently measured Pr to be predominantly trivalent in this compound, so we do not expect significant contributions from more than one Pr valence.

<https://journals.aps.org/prb/pdf/10.1103/PhysRevB.66.052503>

3. If the sample is not detwinned, then the measurements will potentially be measuring H or K or both. The authors should not label the peaks only by K.

We thank the Reviewer for bringing this point to our attention. We agree that it would be more correct not to label the peaks only by K. We have modified the axis labels from “K” to “H or K”, where available, except for places where the label has been left as “K” for the sake of readability. We have modified the main text to make this clear to the reader.

4. An alternate explanation for some aspects of the data, including that the peaks are observed off resonance, may be diffuse scattering from Pr disorder. The nature of the disorder may be such that peak are narrow in L (with little deviation of the Pr positions from their ideal values along the c-axis), but more positional significant disorder within the ab plane. Some mention of this possibility is perhaps warranted in the paper. Measurement of another YBCO variant with other substitution for Y (say Dy) may provide insight into this possibility. Notably, the temperature dependence of the peak, particularly the dip below T_c does not fit neatly into this this description.

This is an excellent idea. The reference below has actually already performed the exact measurement that the Reviewer has proposed – substituting Dy for Y – and did not observe any peaks at the 3D CO wavevector. Additionally, this is not reconcilable with the temperature dependence losing intensity at the onset of superconductivity, as the Reviewer has mentioned. <https://link.aps.org/doi/10.1103/PhysRevB.102.195149>

It is noteworthy that even in the case of $\text{NdBa}_2\text{Cu}_3\text{O}_{6.6}$ there was no 3D CO peak observed. <https://www.science.org/doi/10.1126/science.1223532>

Pr, which is the largest of all rare-earth 3+ ions (except for Ce which does not form the YBCO structure), is unique in the rare-earth family, in terms of how much hybridization overlap there is with the CuO_2 planes. This is also the reason why only Pr suppresses superconductivity. <https://link.springer.com/article/10.1557/JMR.1992.1917>

Finally, as noted to Reviewer #2, x-ray absorption fine structure measurements have shown that the Pr ions are relatively well-ordered at the Y sites but that there is clear disorder in the CuO_2 planes and in the oxygen environment around the Pr, which makes the enhancement of the out-of-plane correlation length far more intriguing and significant. We have added this statement to the main text.

<https://journals.aps.org/prb/abstract/10.1103/PhysRevB.49.3432>

More minor issues

1. For resonant soft x-ray scattering of long-range order, the width of a peak may be determined by the finite penetration depth of the x-rays into the sample rather than representing the correlation length of some order. Evidence that this is the case can be a peak width that varies with photon energy (since the absorption length varies with photon energy). Can the authors compare the peak width to the expected absorption length at the measured photon energies?

We thank the Reviewer for bringing this point to our attention. We would like to acknowledge that a limited penetration depth could only result in broadening our measurement of the 3D CO, implying that the true 3D CO correlation length may be longer than we are reporting.

We have calculated the absorption length for this compound ($\text{Pr}_{0.3}\text{Y}_{0.7}\text{Ba}_2\text{Cu}_3\text{O}_7$), energy (930.3 eV), and angle of incidence (~ 10 degrees) and find that absorption length to be approximately 384 Å. This is very similar to our reported correlation length of 364 Å. Though these two values are of similar magnitude, we do not believe it plays a significant role in limiting the 3D CO correlation length because the (002) reflection has a correlation length that is within the experimental uncertainty of the 3D CO peak, even though it was measured under conditions where the absorption length is roughly an order of magnitude greater ($\sim 3,000$ Å). We believe this is further corroborated by the plot below, showing that the FWHM of the 3D CO peak does not vary significantly across the Pr and Cu absorption edges.

We have modified the main text to reflect this important observation and have added the figure above to the SI.

2. Even off resonance, there should be a significant reduction in scattering intensity for pi polarization relative to sigma polarization for this scattering geometry. As such the presentation of the polarization dependence of the scattering intensity does not necessarily provide an indication that the states are in-plane states. This is associated with the standard polarization factor for x-ray diffraction where scattering for pi polarized photons is zero when the incident and scattered photon polarization are orthogonal. Without further analysis, it is not clear to me that the scattering involves planar orbitals.

We agree with the Reviewer's thoughts and apologize for the confusion. We did not mean to say that the polarization dependence was evidence for planar orbitals, it was only intended to show the similarities between other cuprates. In order to avoid further confusion, we have removed this information entirely as it does not add anything substantial to the main point of the paper.

3. The "XAS" measured in TFY is likely not a good measure of the x-ray absorption. It may be affected by self-absorption corrections at the Pr edge and will also be affected by the difference in quantum efficiency of fluorescence yield for Pr M emission vs. Cu L emission. When edges do not overlap, this can be ignored, but when they do, the FY spectra is unlikely to resemble the x-ray absorption co-efficient. This should be clarified in the text.

This is an important clarification and we thank the Reviewer for bringing it to our attention. The only information we are extracting from the TFY is the location of the Cu L₃ and Pr M₅ peak positions, which we then use to compare with the scattered intensity's dependence on photon energy. We have added text to the SI to clarify this point and to state that it is not a good measure of the x-ray absorption coefficient for the reasons that Reviewer has mentioned.

4. How was the scattering intensity vs energy measured? This is not specified in the paper. Was this via an energy scan at fixed Q? Energy scans at fixed Q can be misleading for sharp peaks that have peak position that varies as a function of photon energy due to the energy dependence of the index of refraction.

These are excellent questions. The scattering intensity vs energy was measured by performing a rocking curve scan at each energy, subtracting the background, fitting the resulting peak, and plotting the areas, specifically to avoid the issue that the Reviewer has mentioned. We have updated the SI to explain this in detail. We thank the Reviewer for bringing this to our attention.

5. In lines 43- 45 , the authors note: "The 2D character of the surviving dx²-y² orbital is observable in both transport and scattering measurements, for example, evidenced by its anisotropic electrical and thermal conductances³ and overwhelmingly broad scattering peaks along L in reciprocal space⁴, respectively."

It is understandable that transport is anisotropic in these materials, due to the dx^2-y^2 character of the states orbitals at the Fermi surface (this is found from first principles DFT calculations), I don't think the relationship between the fact that holes are in dx^2-y^2 orbitals and the correlation length of CDW peaks along the L direction is very clear. The coupling between layers rather than the orbital symmetry within an individual layer may be most relevant for the c-axis correlation length. For instance, ortho ordering peaks in YBCO are broad in L. These peaks, however, principle result from Cu and oxygen in the chain layers, which involve Cu dy^2-z^2 states and O p_z states – states are that are not orthogonal to z. As such, it is coupling between layers rather than the orbital symmetry of an individual layer that may be most relevant.

We thank the Reviewer for raising this point. We agree with the Reviewer's comments entirely and have changed the text accordingly:

The 2D character of the surviving $d_{x^2-y^2}$ orbital is directly observable in transport measurements, evidenced by its anisotropic electrical and thermal conductances, for example. The 2D character of the system is also observable in scattering measurements, where the lack of coupling pathways between adjacent planes can yield overwhelmingly broad scattering peaks along L in reciprocal space.

Reviewer #2 (Remarks to the Author):

The authors present a resonant soft x-ray scattering (RSXS) study of Pr-substituted YBCO and find charge order at a wavevector (0, -0.335, 1). The key and unique finding is an extremely long c-axis correlation length of at least 364 Å. This value far exceeds the correlation lengths of similar charge order reported previously in YBCO perturbed by strong magnetic fields or strain. The mechanism of the strong c-axis correlations is argued to be related to interlayer coupling mediated by the Pr 4f orbitals, which hybridize with the O 2p orbitals in the CuO₂ planes. The findings are striking and clearly presented and demonstrate control of strongly correlated phases through "orbital engineering". I am in favor of publication in Nature Communications although there are a few minor points and questions the authors should address.

1. Did the authors collect data on the temperature dependence of the lineshape? Fig 3c shows plots of intensity vs temperature (I assume this means peak intensity), but if possible, it would be useful to see the dependence of correlation length and/or integrated intensity vs temperature as well.

This is an excellent question. The plot on the left below shows the FWHM (along L) of the 3D CO peak. The width of the 3D CO peak does not appear to vary significantly across the temperature range, which we believe is due to being limited by the coherence length of the lattice. We have added this figure to the SI.

2. How does the in-plane correlation length of the CO compare to that in Refs. 4, 23-27? On first glance it appears the peak in Fig 2b is much wider than that in the previous references. This might be related to the increased chemical disorder due to Pr substitution, in which case the enhancement of correlation length along the c-axis is even more striking.

X-ray absorption fine structure measurements have shown that, while the Pr ions are relatively well-ordered at the Y sites, there is clear disorder in the CuO₂ planes and in the oxygen environment around the Pr. We thank the Reviewer for raising this important point and have added its discussion to the main text.

<https://journals.aps.org/prb/abstract/10.1103/PhysRevB.49.3432>

It might be worth noting this in the text and possibly including in the supplement a comparison similar to Fig 2c, but along K.

This is an excellent idea and we have added the following figure to the SI, which compares the present work against the 3D CO stabilized by epitaxial strain, which offers the most direct comparison in terms of photon energy, scattering geometry, and temperature.

We note that, while this peak is broader in-plane than other observations of 3D CO in YBCO, the hump-like background measured by resonant x-ray scattering is a common feature in cuprates and is widely attributed to fluctuations.

<https://www.science.org/doi/10.1126/science.aav1315>

3. lines 188-189: Single-band models can describe 3d systems, although not Pr-YBCO since there are two CuO₂ layers per unit cell. If I understand the intention of this sentence correctly, it would be more accurate to say "...validity of 2d models to describe layered systems like the cuprates...".

We thank the Reviewer for bringing this point to our attention. We have modified the text accordingly.

4. lines 138-139: "sizeable lattice distortion unseen in other cuprate system". Is this true? Tranquada Nature 375, 561–563 (1995) reported observation of charge order via neutron scattering, due to coupling of the lattice to charge. Certainly the present result is striking, given that this has not been observed previously in YBCO.

This is a good point and we thank the Reviewer for bringing it to our attention. We have revised the text from "unseen" to "rarely seen" and have included a reference to Tranquada et al..

Reviewer #3 (Remarks to the Author):

I have read the manuscript “Stabilization of three-dimensional charge order CO through interplanar orbital hybridization in $\text{Pr}_x\text{Y}_{1-x}\text{Ba}_2\text{Cu}_3\text{O}_{6+\delta}$ ” by Alejandro Ruiz and collaborators submitted to Nature Communications for consideration. The manuscript reports resonant x ray diffraction signatures of charge order which do not commensurate with the real space lattice. Pr substitution triggers a change from 2D charge order with short out of plane correlation (10 Å in underdoped cuprates with similar oxygen content), to 3D charge order with a long out of plane correlation length of 360 Å, which is actually instrument resolution limited. Resonant measurements at the Pr M edge confirm that Pr orbitals play a role in the 3D coupling. DFT +U calculations are used to show that the Pr f_z x^2-y^2 band, crossing the Fermi level, is responsible for the out of plane coupling of the charge ordered states.

This is an important manuscript showing that the 3D CO developing in underdoped cuprates triggered by disorder, strain or large magnetic field is in fact a different form of static order coexisting and competing with superconductivity. Furthermore, this paper adds the important new information that the 2D CO concomitant with the superconducting state in underdoped cuprates is not an essential ingredient in the big picture of the high T_c superconductivity of the cuprates.

An important information missing is how the proposed electronic coupling through the Pr f states affects other electronic states, aside from the CO state, in particular how doping is modified (if it is at all).

We thank the Reviewer for raising this point. From prior calculations cited in the text, it is clear that the effective doping is indeed affected. The Pr f -band above the Fermi level takes holes from the superconducting band, which is consistent with the observation that T_c is suppressed with increasing Pr concentration. We have added this important point to the main text.

<https://journals.aps.org/prl/pdf/10.1103/PhysRevLett.74.1000>

In addition, it is important to know how Pr doping modifies the dimensionality of the superconducting state and the extent superconductivity becomes more 3D as also does the CO state.

It has been shown that the zero-temperature in- and out-of-plane superconducting coherence lengths increase monotonically with Pr concentration and are substantially longer than in pristine YBCO and has been attributed to increased coupling between CuO_2 planes through the bonding with substituted Pr ions. This is an interesting result that suggests there is a relationship between

CO and SC, both of which become more 3D with increasing Pr concentration, and we have added it to our discussion in the main text.

<https://journals.aps.org/prb/abstract/10.1103/PhysRevB.45.10609>

In addition, it would be very important to have information on the evolution of the out of plane correlation length of the CO state with doping. I.e., if hole doping of the Pr doped compound (by oxygen depletion for example) would restore the 2D picture.

This is an excellent idea. In fact, we are currently working on securing funding for these experiments. We would like to study the 3D CO as a function of i) Pr concentration, ii) O concentration, and iii) counter-doping via substitution of Ca^{2+} at the Y^{3+} sites. We hope the Reviewer will understand the logistical details of the ensuing process.

REVIEWER COMMENTS

Reviewer #1 (Remarks to the Author):

The authors have revised their manuscript. On the whole, I think the revised version is an improvement over the original. The primary result a CDW order peaked at integer L values and with longer c-axis correlation lengths has been observed in YBCO before. The novelty here is the apparent observation of the order in single crystal that is unstrained and not under an applied magnetic field. Moreover, the observed Bragg peak is much narrower in L than in H or K, indicating a longer c-axis correlation length than the in plane correlation length (such a density wave order is seemingly not consistent with naïve conceptions of density wave order being driven by in-plane correlations, with weaker c-axis coupling.)

Despite these interesting observations, I still do not recommend publication in Nature Communications. My reasons for this assessment are twofold. 1. I think the interest/impact of this result falls short of the standard for publication in a premier journal such as Nature Communications. 2. The manuscript remains too ambiguous and unclear. does not clearly address some of the concerns from my first review.

Regarding interest and impact, I think the overall impact of this result and the lessons that can be learned from it limited by the peculiarity of Pr doping in YBCO. Because Pr states hybridize strongly with the CuO₂ planes, it is sufficiently different from the cuprate paradigm to shed great insight into key outstanding questions regarding CDW order in the cuprates, including why 3D-like order forms under strain or applied field. Moreover, this think there remain significant limitations of the measurements and analysis that limit the confidence with which one can draw firm conclusions.

Specifically,

1. The lack of clear data about the H or K dependence of the CDW peaks hampers conclusions. From what I gather, a cut through H at constant L dependence was only measured at 50 K. It is important to understand if this remains a peak at high temperatures. Moreover, the lineshape along H is unusual – not the Lorentzian like lineshape that is measured for CDW peaks in other cuprates. Coupled with the fact that the correlation length in L is seemingly much longer than H (a result that is very difficult to understand given theories of CDW order in the cuprates), it remains plausible to me that there are additional contributions to the scattering that is not directly related to the CDW order observed in other cuprates.

2. I still think the authors are oversimplifying the energy dependence of the scattering. Given the energetic overlap of the Pr M and Cu L edges, the energy dependence of the scattering is unavoidably complex. At best one can state that Pr atoms must contribute to the scattering and that Cu atoms most likely do. As the authors note, it is logical given the hybridization of the Pr and Cu states, as well as the T dependence, which indicates a coupling to superconductivity, that both Cu and Pr states contribute. However, beyond that assertion, the energy dependence is unavoidably complex. One can speculate that the energies correspond to different resonances. However, at the Cu L edge, it would be perfectly reasonable to still have scattering coming from the Pr M edge. Apriori, there is no reason why the energy dependence of the scattering at the Pr M edge ought to give a single Lorentzian-like peak, regardless of whether there are different Pr valences present. Multi-peak structures in the energy dependence of RE M edge scattering are in fact quite common. Moreover, at intermediate energies, resonant scattering from both Pr and Cu sites would contribute the scattering.

It think the authors should acknowledge this more clearly in the main text and include simply the energy of the scattering, without denoting the energy being associated with Cu L vs. Nd M. and The paper still includes much of the discussion and figures attributing scattering at 930 eV to Cu L and lower energy to Pr M. Given the overlap in the edges, I don't think these can be distinguished.

Minor concerns:

- Figure S3 should be removed. As per my earlier comments, the narrow peak in H observed here is misleading. It is highly likely to be misinterpreted by readers that casually scan the figures without reading the full text of the paper.
- The authors state that the peaks are “resolution-limited by the width of the crystallographic Bragg peaks” Resolution limited conventionally refers to being limited by the experimental resolution. Is this the case here? It is unclear. Instrumental resolution can be from the energy resolution and angular resolution of the instrument, which are different for different Bragg peaks. Having a correlation length that is equal to the correlation length of a structural Bragg peak is something else.

Reviewer #2 (Remarks to the Author):

The authors have addressed all of the minor points raised in the previous round of review and have revised and improved their manuscript accordingly. I support publication in Nature Communications.

Reviewer #3 (Remarks to the Author):

I have read the manuscript "Stabilization of three-dimensional charge order CO through interplanar orbital hybridization in $\text{Pr}_x\text{Y}_{1-x}\text{Ba}_2\text{Cu}_3\text{O}_{6+\delta}$ " by Alejandro Ruiz and collaborators and author's response to the criticism in the first review round.

Authors have satisfactorily addressed my criticism and made changes to the manuscript according to it.

To my question on the possible doping effect of Pr substitution (besides changing the dimensionality of charge order) authors have made clear that there is indeed a doping effect associated to hole trapping by the Pr f-states. Authors have made citation to theoretical calculations supporting this picture.

To my question on whether upon Pr substitution superconductivity becomes more 3D as also does the CO state, authors have clarified that the zero-temperature in- and out-of-plane superconducting coherence lengths increase also with Pr concentration what suggests a relationship between charge order and superconductivity.

I maintain my view that this is an important manuscript which advances in the knowledge of the interplay between charge ordered states and superconductivity in underdoped cuprates. The paper should be of interest in the wide Materials Physics and Chemistry and Condensed Matter Physics as high T_c superconductivity continues to be a very important open problem. My recommendation is to accept this manuscript in its present revised form in Nature Communications.

Reviewer #1 (Remarks to the Author):

The authors have revised their manuscript. On the whole, I think the revised version is an improvement over the original. The primary result a CDW order peaked at integer L values and with longer c-axis correlation lengths has been observed in YBCO before. The novelty here is the apparent observation of the order in single crystal that is unstrained and not under an applied magnetic field. Moreover, the observed Bragg peak is much narrower in L than in H or K, indicating a longer c-axis correlation length than the in plane correlation length (such a density wave order is seemingly not consistent with naïve conceptions of density wave order being driven by in-plane correlations, with weaker c-axis coupling.)

Despite these interesting observations, I still do not recommend publication in Nature Communications. My reasons for this assessment are twofold. 1. I think the interest/impact of this result falls short of the standard for publication in a premier journal such as Nature Communications. 2. The manuscript remains too ambiguous and unclear. does not clearly address some of the concerns from my first review.

Regarding interest and impact, I think the overall impact of this result and the lessons that can be learned from it limited by the peculiarity of Pr doping in YBCO. Because Pr states hybridize strongly with the CuO₂ planes, it is sufficiently different from the cuprate paradigm to shed great insight into key outstanding questions regarding CDW order in the cuprates, including why 3D-like order forms under strain or applied field. Moreover, this think there remain significant limitations of the measurements and analysis that limit the confidence with which one can draw firm conclusions.

We would like to point out that 3D charge ordering stabilized by Pr substitution is no more peculiar to YBCO than that induced by strain or applied field, as 3D CO has not yet been observed in any other cuprate. Whether 3D CO is unique to YBCO remains a key outstanding question regarding CO in cuprates; the method of hybridization-induced 3D CO stabilization via substitution of system-appropriate elements may offer a unique and accessible route towards answering this question as it is not subject to the restrictions incurred by the applications of strain or magnetic fields that preclude many measurements.

Specifically,

1. The lack of clear data about the H or K dependence of the CDW peaks hampers conclusions. From what I gather, a cut through H at constant L dependence was only measured at 50 K. It is important to understand if this remains a peak at high temperatures. Moreover, the lineshape along H is unusual – not the Lorentzian like lineshape that is measured for CDW peaks in other cuprates. Coupled with the fact that the correlation length in L is seemingly much longer than H (a result that is very difficult to understand given theories of CDW order in the cuprates), it remains plausible to me that there are additional contributions to the scattering that is not directly related to the CDW order observed in other cuprates.

We have modified the text to reflect that it is possible that there are additional contributions to the peak lineshape in addition to the charge ordering contribution to the intensity (line 173 in manuscript PDF). This does not significantly detract from the message of the manuscript as follows: the temperature and

reciprocal space dependence demonstrate that 3D charge ordering is at least a major component of the measured scattering and that it still competes with superconductivity, which has been the primary motivation behind cuprate charge ordering research.

2. I still think the authors are oversimplifying the energy dependence of the scattering. Given the energetic overlap of the Pr M and Cu L edges, the energy dependence of the scattering is unavoidably complex. At best one can state that Pr atoms must contribute to the scattering and that Cu atoms most likely do. As the authors note, it is logical given the hybridization of the Pr and Cu states, as well as the T dependence, which indicates a coupling to superconductivity, that both Cu and Pr states contribute. However, beyond that assertion, the energy dependence is unavoidably complex. One can speculate that the energies correspond to different resonances. However, at the Cu L edge, it would be perfectly reasonable to still have scattering coming from the Pr M edge. Apriori, there is no reason why the energy dependence of the scattering at the Pr M edge ought to give a single Lorentzian-like peak, regardless of whether there are different Pr valences present. Multi-peak structures in the energy dependence of RE M edge scattering are in fact quite common. Moreover, at intermediate energies, resonant scattering from both Pr and Cu sites would contribute the scattering. It think the authors should acknowledge this more clearly in the main text and include simply the energy of the scattering, without denoting the energy being associated with Cu L vs. Nd M. and The paper still includes much of the discussion and figures attributing scattering at 930 eV to Cu L and lower energy to Pr M. Given the overlap in the edges, I don't think these can be distinguished.

We have changed the labels of the two features in the energy dependence from the Pr and Cu resonances to simply peaks A and B and have modified their references throughout the text accordingly. We have modified the text to state clearly that we cannot precisely determine the origin of these two features due to the unavoidable complexity resulting from the energetic overlap of the Pr and Cu edges .

However, we continue to maintain the belief that the two features in the energy dependence being associated with the Pr and Cu edges observed in the absorption spectra is overwhelmingly the most likely scenario for the following reasons: i) the two features in the energy dependence lie on top of the Pr and Cu edges observed in the absorption spectra; ii) comparison to other studies of pristine and RE-substituted YBCO¹ suggests that Pr must contribute, while the behavior of the temperature dependence suggests that Cu must contribute, iii) it is logical given the hybridization between Pr and the planar electronic states that both Cu and Pr contribute, iv) there is nothing to suggest that this is not the case, aside from it being a possibility. We have stated this belief in the text and have added its supporting arguments to the supplemental information. Nonetheless, we have been careful to ensure that it is clear to the reader that this cannot technically be distinguished with absolute certainty due to the overlapping edges.

Minor concerns:

¹ <https://journals.aps.org/prb/abstract/10.1103/PhysRevB.102.195149>

- Figure S3 should be removed. As per my earlier comments, the narrow peak in H observed here is misleading. It is highly likely to be misinterpreted by readers that casually scan the figures without reading the full text of the paper.

We agree that this figure has a high potential for misinterpretation from casual readers and have removed it entirely.

- The authors state that the peaks are “resolution-limited by the width of the crystallographic Bragg peaks” Resolution limited conventionally refers to being limited by the experimental resolution. Is this the case here? It is unclear. Instrumental resolution can be from the energy resolution and angular resolution of the instrument, which are different for different Bragg peaks. Having a correlation length that is equal to the correlation length of a structural Bragg peak is something else.

We did not mean to refer to experimental resolution in this sentence and have removed the phrase to avoid further confusion.